# One single drug-coated balloon for all shapes/ diameters? Neointimal proliferation inhibition in porcine peripheral arteries

Stephanie Bienek[1], Maciej Kusmierczuk[1], Beatrix Schnorr[2], Ole Gemeinhardt[2], Stephanie Bettink[3], Bruno Scheller[3]*

1 InnoRa GmbH, Berlin, Germany, 2 Department of Radiology, Charité - Universitätsmedizin Berlin, Corporate Member of Freie Universität Berlin and Humboldt-Universität zu Berlin, Berlin, Germany, 3 Clinical and Experimental Interventional Cardiology, University of Saarland, Homburg, Saar, Germany

* bruno.scheller@uks.eu

**Data Availability Statement:** All relevant data are within the paper and its Supporting information files.

## Abstract

### Background

Long diseased vessel segments of peripheral arteries may display irregular shapes with different diameters. The aim of this study was to investigate inhibition of neointimal proliferation in porcine peripheral vessels with different diameters covered by one single hyper-compliant drug-coated balloon (HCDCB), compared to conventional drug-coated balloons (DCB), each selected according to the respective vessel diameter.

### Methods and results

Neointimal proliferation was stimulated in proximal and distal segments of the peripheral arteries by balloon overstretch and stent implantation. Inhibition of neointimal proliferation by one single HCDCB was compared to two vessel diameter-adjusted DCB per artery and to one single uncoated hyper-compliant balloon (HCB). Sixteen HCB, 16 HCDCB, and 32 DCB were used in 16 arteries each. Quantitative angiography (QA), optical coherence tomography (OCT) and histology showed a similar anti-restenotic effect for one HCDCB compared to two vessel diameter-adjusted DCB in narrow distal and wider proximal segments (QA diameter stenosis: 18.7±12.3% vs. 22.8±15.5%, p = 0.535; OCT area stenosis: 21.4±11.6% vs. 23.6±12.3%, p = 0.850; histomorphometry diameter stenosis: 27.5±7.1% vs. 26.9±8.0%, p = 0.952) and indicated significant inhibition of neointimal proliferation by HCDCB vs. uncoated HCB (QA diameter stenosis: 18.7±12.3% vs. 30.3±16.7%, p = 0.008; OCT area stenosis: 21.4±11.6% vs. 34.7±16.0%, p = 0.004; histomorphometry diameter stenosis: 27.5±7.1% vs. 32.5±8.5%, p = 0.038).

### Conclusions

HCDCB were found to be similar effective as DCB in inhibiting neointimal proliferation in vessel segments with different diameters. One single long HCDCB may allow for treatment of segments with variable diameters, and thus, replace the use of several vessel diameter-adjusted DCB.

**Funding:** the author(s) received no specific funding for this work.

**Competing interests:** Bruno Scheller is a shareholder of InnoRa GmbH. This does not alter our adherence to PLOS ONE policies on sharing data and materials.

## Introduction

Drug-eluting stents (DES) and drug-coated balloons (DCB) have shown to inhibit excessive neointimal growth after vessel wall trauma induced by mechanical revascularization techniques [1–5]. DCB differ from DES by the rapid drug delivery from a uniform support structure, and in leaving no permanent implant in place. DCB allow deployment at vascular sites which may not be reached by stents or where stent implantation may result in unfavorable outcome [6–8]. Contrary to self-expanding stents the common non-compliant and semi-compliant balloons do not adjust to the vessel lumen diameter and shape but are inflated to a predetermined cylinder. To obtain the required vessel wall contact the diameter of the balloon has to be chosen in advance according to the desired vessel diameter of the treated segment. The entrance of side branches may be compressed. Steps and changes in the diameter of the treated segment cause problems. In long lesions in e.g. superficial femoral artery (SFA), one of the longest vessels in the body reaching up to 30 cm in length, the lumen diameter may change from wide to narrow. Therefore, for optimal treatment of stenotic arteries, careful selection of DCB according to the respective vessel diameters is necessary to avoid insufficient treatment of lesions due to suboptimal drug transfer to the vessel wall by undersized DCB or undesired lumen expansion and vessel wall injury/perforation by oversized DCB [9].

Hyper-compliant balloons were shown to comprise some favourable features such as (i) coverage of long lesions in peripheral arteries due to their long length, (ii) adjustment to different vessel diameters especially in long arterial segments by displaying perfect conformability to the vessel anatomy, and (iii) high drug transfer at low pressure if further dilatation is not required [10]. It was also demonstrated in our previous animal study [10] that hyper-compliant balloons coated with paclitaxel (HCDCB) inhibit neointimal proliferation in a single segment of internal iliac arteries after experimental vessel overstretch caused by angioplasty and stent implantation. Comparison with standard DCB (coated with paclitaxel) demonstrated that both HCDCB and DCB are similarly effective in inhibiting neointimal proliferation, while treatment with uncoated hyper-compliant balloons (HCB) is less efficacious.

The aim of the present study in porcine peripheral arteries was to compare the inhibitory effect of one single hyper-compliant paclitaxel-coated balloon on neointimal proliferation in vessel segments with different diameters with that of two commercial DCB that were selected according to the diameter of the vessel. Comparison was also made with uncoated hyper-compliant balloons (HCB).

## Materials and methods

### Balloon catheters and drug content

Hyper-compliant balloon catheters, uncoated (HCB) and paclitaxel-coated (HCDCB), consisted of a balloon which was made of a stretchable biocompatible elastomer and fixed on a catheter shaft compatible to 0.035" guidewires, in an over-the-wire version. HCDCB with balloon lengths of 100 mm or 150 mm were loaded with paclitaxel at a dose of ca. 50–65 μg/mm of the balloon length (targeting at 3 μg paclitaxel/mm$^2$ of the balloon surface at 7 mm diameter) and an excipient. Commercial DCB were coated with 3.5 μg paclitaxel/mm$^2$ of the balloon surface (In.PACT Admiral, Medtronic, Minneapolis, MN, USA), 4.0, 5.0, or 6.0 mm diameter and 40 mm length. Uncoated and coated hyper-compliant balloons were sterilized by ethylene oxide.

### Stents

Different types and sizes of uncoated balloon-expandable bare metal stents were used to induce overstretch and neointimal proliferation in proximal and distal segments of internal

iliac and femoral arteries—for proximal femoral segments: 6.0x16 mm Perico stents (Panmed US, Largo, FL, USA), for distal femoral segments: 4.0x19 mm Express Vascular SD Pre-mounted stent system (Boston Scientific, Maple Grove, MN, USA), for proximal segments of the internal iliac artery: 5.0x12 mm Perico stents, and for distal segments of the internal iliac artery: 3.5x16 mm REBEL Monorail PtCr stent system (Boston Scientific).

## Study design

The study design of the efficacy study is presented in Fig 1. A total of 12 domestic pigs were enrolled, and in total four arteries were treated per pig. Each femoral and internal iliac artery of the animals was allocated to treatment with HCDCB (one per artery), uncoated HCB (one per artery), or commercial DCB (two per artery) in such a way that the total number of vessels was the same for the three treatment groups (n = 16 for each treatment group). The course of the study is described in more detail below.

## Animals

The study used the established porcine model. Swine are the appropriate model for the study of neointimal hyperplasia and for anti-restenosis therapies based on anatomical, physiological and hematological similarities to humans. It was performed in castrated male domestic pigs (body weight 22.9±1.4 kg) at the Institute of Medical Technology and Research (IMTR GmbH, Rottmersleben, Germany). All animals received standard care outlined in accordance with the EU Commission Directive 86/609/EEC and the German Animal Protection Act based upon the Animal Ethics Committee approvals (Saxony-Anhalt, Germany). Before the study start a health diagnosis for each animal was made, and only healthy animals were enrolled.

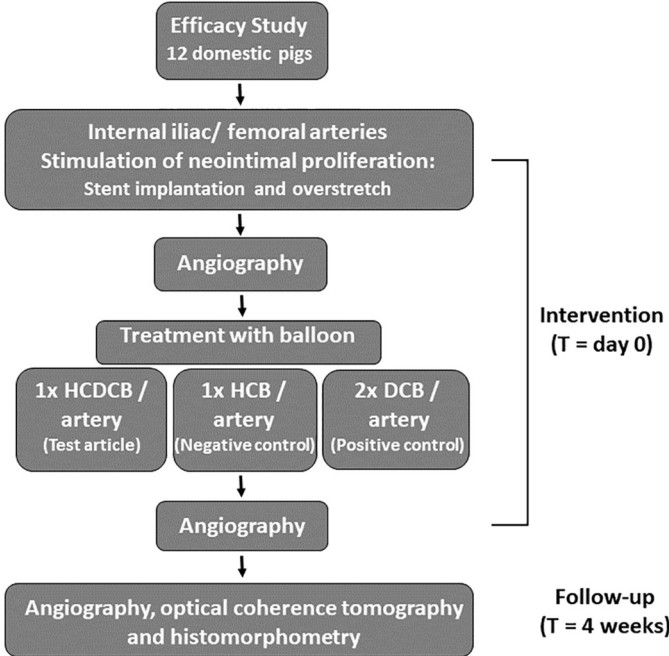

**Fig 1. Flow chart describing the study design including follow-up.** HCDCB = hyper-compliant drug (paclitaxel)-coated balloon; HCB: uncoated hyper-compliant balloon; DCB: drug (paclitaxel)-coated PTA balloon.

## Treatment

**Anesthesia and pre-interventional procedure.** Clopidogrel (75 mg) and acetylsalicylic acid (100 mg) per day were administered starting 2 days before the procedure and continued until completion of the study. Verapamil hydrochloride (120 mg) was given within 24 hours prior to the procedure to reduce vascular spasm during the procedure.

The pigs were sedated by intramuscular injection with ketamine (Ursotamin®, 20 mg/kg body weight, Serumwerk Bernburg, Germany) and xylazine hydrochloride (Xylazin®, Riemser Arzneimittel GmbH, 2 mg/kg body weight, Greifswald, Germany). Anesthesia was induced with propofol (Recofol 1%, 3 mg/kg body weight, Curamed Pharma GmbH, Freiburg, Germany) followed by intramuscular administration of meloxicam (Metacam®, 0.4 ml/10 kg body weight, Boehringer Ingelheim Vet Medica, Ingelheim, Germany) for post operative pain management and oxytetracycline hydrochloride (Ursocyclin® 1 ml/5 kg body weight, Serumwerk Bernburg) for prophylactic antibiosis. Also as analgetic, butorphanol (Morphasol® 10 mg/ml, 0.1 ml/kg body weight, aniMedica international GmbH, Frankfurt, Germany) was intravenously injected. The animals were orotracheally incubated (Endonorm, Rüsch GmbH, Böblingen, Germany) and ventilation was started with a mixture of 30–60 vol% of pure oxygen, 40–70 vol% air and 1–2 vol% of isoflurane (Isofluran Curamed, Curamed).

Access for a 7F peripheral guiding sheath (Terumo Destination, 65 cm in length, Terumo Europe, Belgium) was provided through an external carotid artery. Blood pressure was recorded once before and once 2–5 min after the treatment. Heparin-Natrium 5000 IU (Heparin, B.Braun Melsungen AG, Melsungen Germany) and 250 mg DL-lysine mono(acetylsalicylate) (Aspisol®, Bayer AG, Germany) were administered intraarterialy as a bolus. Under fluoroscopic control the 7F guiding sheath was introduced directly (i.e., without an arterial sheath) into the aorta. Angiography of femoral and internal iliac arteries was performed before and after treatment using a Siemens AXIOM Artis zee fluoroscope. The arteries were visualized using iopromide (Ultravist®-370, BSP AG, Berlin, Germany) as contrast agent. Throughout the procedure, the electrocardiogram, oxygenation and temperature were monitored continuously.

**Interventional procedure.** Balloon expandable bare metal stents were introduced over a 0.014" guide wire (Galeo Pro F, 190 or 300 cm, Biotronik, Berlin, Germany) and implanted in proximal and distal segments of each right and left femoral and internal iliac artery. Stent implantation was done consecutively (i.e., distal vessel segment first and shortly thereafter proximal vessel segment). Location of stents in arteries is depicted in Fig 2. Inflation pressure was applied to achieve ca. 10–20% overstretch of the vessel diameter (Table 1).

Balloons (HCB, DCHCB and DCB (In.Pact Admiral)) were introduced over the same guide wire as used for stents. They were pushed through the valve of the 7F guiding sheath (Terumo) with the introducer (Adelante®, Oscor, Palm Harbor, FL, USA) keeping the valve open to protect the coating. In case of treatment with an HCB or an HCDCB (for both devices: 100 mm and 150 mm balloon length for internal iliac and femoral arteries, respectively) the balloon was advanced to and placed in the artery in such a way that it simultaneously and completely covered both proximal and distal stented segments. The balloon was inflated at the treatment site by applying minimal pressure allowing for expansion in its full length (Table 2).

For DCB (In.Pact Admiral) each of both stented vessel segments in the femoral and internal iliac artery was separately treated with a DCB selected according to the diameter of the vessel lumen. The sequence of treatment with a DCB in femoral and internal iliac arteries was as follows: first with a 4.0x40 mm balloon (femoral or internal iliac) in the distal segment followed by 6.0x40 mm (femoral) or 5.0x40 mm (internal iliac) balloon in the proximal segment. DCB were inflated at nominal pressure of 6 atm (internal iliac arteries) or 8 atm (femoral arteries).

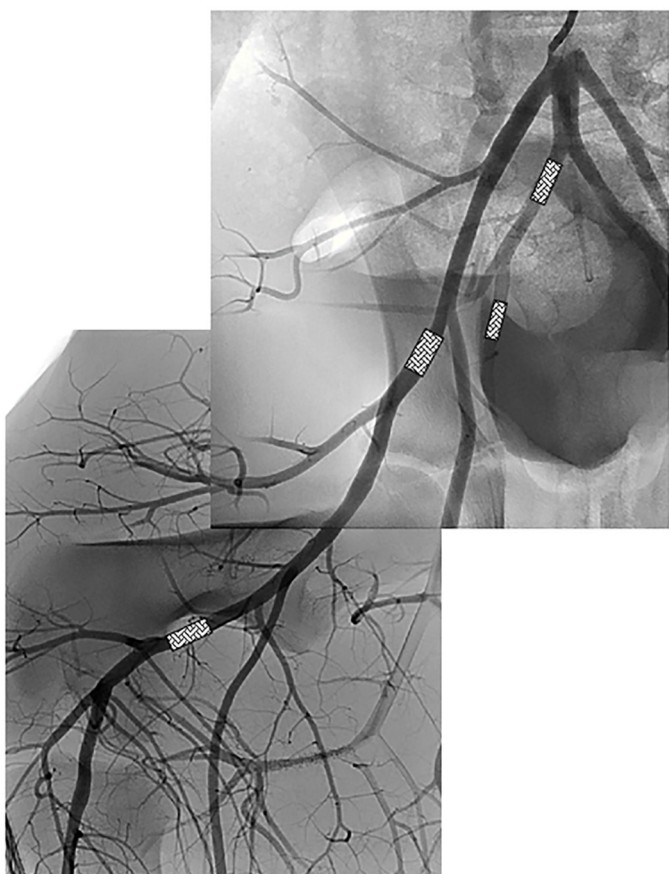

**Fig 2. Location of stent implantation in vessel segments of femoral and internal iliac arteries.** Two stents (depicted here as box) per artery were implanted—one each in the proximal and distal vessel segment. Dimensions of the stents are described in the Materials and methods section.

**Table 1. Inflation pressure applied for stent implantation in porcine peripheral arteries.**

| Artery | Vessel segment* | Inflation pressure [atm] |
|---|---|---|
| **femoral** | proximal | 10.6±2.3 |
| | distal | 14.3±1.7 |
| **internal iliac** | proximal | 9.8±2.7 |
| | distal | 13.8±2.0 |

* n = 24 for each vessel segment.

**Table 2. Inflation filling volume and inflation pressure applied for hyper-compliant balloons.**

| Artery | Balloon length [mm] | Inflation filling volume [ml] | | Inflation pressure [atm] | |
|---|---|---|---|---|---|
| | | HCDCB | HCB | HCDCB | HCB |
| femoral | 150 | 4.50±0.46 | 4.38±0.88 | 1.14±0.40 | 0.55±0.45 |
| internal iliac | 100 | 2.13±0.23 | 2.06±0.32 | 1.23±0.70 | 0.50±0.43 |

HCDCB, hyper-compliant paclitaxel-coated balloon, HCB, uncoated hyper-compliant balloon. n = 8 for each artery.

Inflation time for all balloons was 120 sec. The balloons were collected for residual drug extraction and quantification.

At 4-week follow-up angiography followed by optical coherence tomography was performed. Optical coherence tomography (OCT) images of stented vessel segments were acquired using the ILUMIEN™OPTIS™ system (Abbott Medical GmbH, Wetzlar, Germany). A Dragonfly™ OPTIS™ imaging catheter (Abbott) was advanced via a 7F introducer (Avanti+, Cordis, Miami Lakes, FL, USA) to the vessel segment via a 6F guiding catheter (Launcher JL3.5, Medtronic, Minneapolis, MN, USA) and 0.014" guide wire. Images of the vessel segment were acquired through a single injection of contrast agent (Ultravist®-370) using power injector and automated OCT pullback at a rate of 18 mm/s and with a pullback length of about 50 mm. Measurements by OCT, however, were not possible in proximal stented segments of femoral arteries because their vessel diameter surpasses the capability of the instrument.

After euthanasia of the animals in deep anesthesia using 10 ml supersaturated potassium chloride (25%, Roland Apotheke, Haldensleben, Germany). Left and right internal iliac and femoral arteries were flushed in direction proximal to distal with physiologic saline solution as a unit *in situ* using an irrigation cannula. The flushed arteries were separated from each other and individually labelled at their proximal end with a surgical thread. Thereafter, the arteries including two stented and balloon-treated vessel segments in each artery were harvested and preserved in 4% formalin solution (Carl Roth GmbH + Co. KG, Karlsruhe, Germany) for histomorphometry.

**Quantitative angiography.** The CAAS II System (Pie Medical Imaging, Maastricht, the Netherlands) was used for quantitative analysis of angiograms. The parameters were evaluated: (i) reference vessel diameter at implantation (baseline) = minimal lumen diameter (MLD) before treatment at the site of subsequent stent implantation ($MLD_{before\ treatment}$), (ii) in-stent minimal lumen diameter after treatment ($MLD_{after\ treatment}$, baseline), (iii) overstretch ratio after treatment (baseline), (iv) in-stent minimal lumen diameter at 4-week follow-up (MLD at FU), (v) reduction in minimal lumen diameter at 4-week follow-up = $MLD_{after\ treatment}$—$MLD_{FU}$ (equal to late lumen loss (LLL) at 4-week follow-up), and (vi) in-stent diameter stenosis at 4-week follow-up: $MLD_{FU}$ in % of $MLD_{after\ treatment}$.

**Optical coherence tomography.** Qualitative and quantitative analyses of images of stented vessel segments acquired by OCT were done with the ILUMIEN™OPTIS™ offline review workstation. The data were obtained at 4-week follow-up from data acquisition of a section which was located in the center of the stented vessel. The parameters were evaluated on cross-sections: (i) lumen area = total area of the inner open space of the vessel within the outer intima surface, (ii) stent inner area = total area of the inner space of a vessel within the stent contour, (iii) neointimal area = stent inner area—lumen area, (iv) stenosis area (%) = [1-(lumen area/inner stent area)] x100, and (v) neointimal thickness = stent inner diameter (mean)-lumen diameter (mean).

**Histomorphometry.** The target segments of the untreated and paclitaxel-treated vessel segments were carefully harvested, and tissue samples for histology were obtained. The parts were embedded in methyl—methacrylate (Heraeus Kulzer, Wehrheim, Germany). Three cross-sections within the stented area (a,b,c), and two sections proximal to the stent and one section distal to the stent (= segments without stent) were cut with a coping saw, polished, and glued on acrylic plastic slides. Final section thickness was between 10 and 20 μm. Sections (a) and (b) were stained by hematoxylin—eosin technique, and the section (c) was stained by Masson-Goldner. After digitalizing, quantitative histomorphometric measurements of these three sections (a,b,c) were taken using the 'NIS BR 3.0' image program (Nikon Europe, Düsseldorf, Germany) and means of the 3 sections were reported. External and internal elastic lamina (EEL and IEL) were marked. The evaluated parameters were: (i) vessel diameter at follow-up

(4-week) histomorphometry (= mean of the longest axis and the longest perpendicular axis), (ii) lumen diameter at follow-up (4-week) histomorphometry (= mean of the longest axis and the longest perpendicular axis), (iii) maximal neointimal thickness at follow-up (4-week) histomorphometry (including media), (iv) vessel area at follow-up (4-week) histomorphometry (= area within EEL), (v) lumen area at follow-up (4-week) histomorphometry (= area within IEL), (vi) neointimal area at follow-up (4-week) histomorphometry (= area between EEL and IEL), (vii) diameter stenosis at follow-up (4-week) histomorphometry = [1-(lumen diameter/ vessel diameter)]x100, and (viii) stenosis area at follow-up (4-week) histomorphometry = [1-(lumen area /vessel area)]x 100. Injury scores were assigned as previously described by Schwartz et al. [11], and the inflammation score for each individual strut was graded as described by Kornowski et al. [12]. The injury score and the inflammatory score for each cross section were calculated as the sum of the individual injury and inflammatory scores, divided by the number of struts in the examined section.

**Quantification of paclitaxel.** For analysis of paclitaxel each balloon was placed in a vial and cut off from the catheter shaft, and a defined volume of acetonitrile was added. The vials were firmly closed and intensely shaken on a vortexer for at least 30 seconds followed by treatment in a ultrasound bath for 30 minutes and centrifugation for 10 minutes (16,000 x g Eppendorf microcentrifuge 5415C). Samples with a high concentration of analyte were diluted with acetonitrile.

Paclitaxel was quantified by HPLC with UV detection (Shimadzu Nexera-i Lc-2040c, 3D, Shimadzu Corporation, Kyoto, Japan). Column: C18, 5 μm, 25 cm x 4.6 mm. Mobile phase: 45% phosphate buffer 0.005 M and 55% acetonitrile, 1 ml/min. Detection: 230 nm. Column temperature: 35˚C. At least 20 μl of the supernatant was injected into the HPLC unit. A standard solution (paclitaxel) was injected during the same run (concentration approximately 50 μg/ml ± 5%).

**Statistical analysis.** Histomorphometric variables of the three cross-sectional planes and OCT variables of two cross diameters within the stent were averaged to obtain a mean value per treated segment. Missing values were not imputed. All data recorded or calculated are displayed as mean ±SD. Displayed p-values represent differences between pairs of data of two balloon groups (HCDCB vs. HCB, or HCDCB vs. DCB) or among three balloons groups (HCDCB, HCB and DCB) and are obtained with the Student's t-test (2-tailed) or one-way analysis of variance (ANOVA) followed by Tukey's or Dunn's multiple comparison test. A P-value of <0.05 was regarded as statistically significant. Calculations were performed using the GraphPad Prism 9.4 software system (GraphPad Software, Inc., San Diego, CA, USA).

## Results

### Paclitaxel analysis of used balloons

After inflation in peripheral arteries, residual paclitaxel on HCDCB was found to be 30.9 ±20.0% (n = 16) of the initial dose and was quite similar to that with 32.5±12.6% for used DCB (n = 31, p = 0.738).

### Device function, treatment and tolerance

A total of 96 stents were implanted in the peripheral arteries of 12 pigs. While all HCDCB were completely advanced into both proximal and distal segments of femoral and internal iliac arteries, two balloons of the HCB and one of the DCB group could not be advanced into the distal segment of the internal iliac (HCB only) and femoral artery (HCB and DCB). These events were not related to the balloons. Proximal and distal stented segments were simultaneously and fully covered by one single long HCB or HCDCB (Fig 3A and 3B). Owing to high

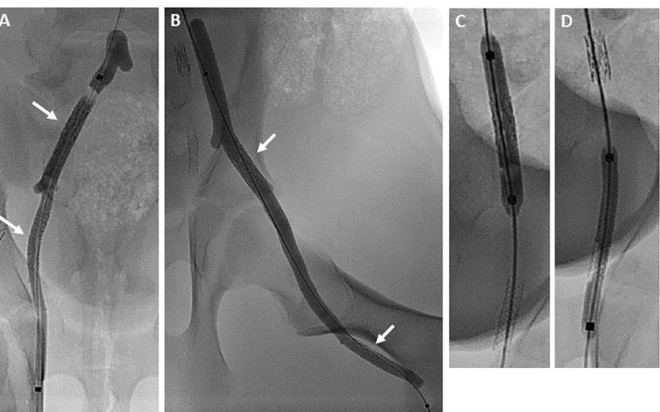

**Fig 3. Inflation of uncoated and hyper-compliant paclitaxel-coated balloon, and conventional paclitaxel-coated PTA balloon in porcine peripheral arteries.** (A and B) One single long hyper-compliant balloon simultaneously covered the proximal and distal stented segment of an artery (white arrow); (A) 100 mm and (B) 150 mm balloon in the internal iliac and femoral artery, respectively. (C and D) By contrast, conventional drug-coated PTA balloons (DCB) were selected according to the diameter of the segment subjected to balloon treatment; (C) 5.0x40 mm and (D) 4.0x40 mm balloon inflated in the proximal and distal segment of the internal iliac artery, respectively.

conformability the balloons adapted well to the vessel structure and stent shape with formation of protrusions at vessel branches. Commercial DCB formed a cylindrical structure with different defined diameters in the stented vessel segments (Fig 3C and 3D). All hyper-compliant balloons and DCB had contact to the vessel wall and stent struts along their entire length.

During the intervention one outflow obstruction possibly due to guidewire-induced dissection between both stented vessel segments in a DCB-treated internal iliac artery was noticed. This was still seen at 4-week follow-up, but was, however, without clinical significance. Otherwise, no signs of thrombotic or embolic events during or after intervention were observed. Vessel spasms proximal and/or distal to the stented vessel segment known to occur after overdilatation were frequently observed (mostly in distal femoral arteries after stent implantation and balloon treatment being graded as moderate to strong). In comparison, only few spasms were seen in internal iliac arteries which were graded as slight. All spasms were temporary since they were not seen at 4-week follow-up.

All twelve animals survived the interventional procedure with weight gain (4.4±1.7 kg) and without recognizable acute or persistent or late signs of intolerance. During the follow-up period, none of the animals experienced clinically obvious disease or showed unusual behaviour indicating pain or weakness.

## Comparison of efficacy of HCDCB and DCB in inhibiting neointimal proliferation

In all three treatment groups (HCB, HCDCB and DCB) stent implantation followed by balloon treatment resulted in overstretch of the stented vessel segment of 15–20% and was not statistically different among the groups (Tables 3 and 4). When considering each stented segment group (proximal and distal) of femoral and internal iliac arteries, quantitative angiography (QA) revealed no relevant differences in angiographic baseline data (i.e., no statistical differences in MLD before and after treatment), except for the proximal segment of femoral arteries where MLD at post treatment was significantly lower in the HCDCB-treated group vs. HCB- or DCB-treated group. This was due to the smaller reference diameter (= $MLD_{before}$) of the vessel segments in the HCDCB-treated group.

**Table 3. Quantitative angiography (QA) of each vessel segment of femoral arteries before and after intervention.**

| | Hyper-compliant balloon | | Standard PTA balloon | p-value | Hyper-compliant balloon | | Standard PTA balloon | p-value |
|---|---|---|---|---|---|---|---|---|
| | Uncoated (HCB) | Coated (HCDCB) | Coated (DCB) | | Uncoated (HCB) | Coated (HCDCB) | Coated (DCB) | |
| | **Femoral proximal** | | | | **Femoral distal** | | | |
| n (analyzed vessels) | 8 | 8 | 8 | | 7* | 8 | 7* | |
| $MLD_{before}$ [mm] | 4.23±0.39 | 3.74±0.62 | 4.05±0.31 | 0.119 | 2.95±0.42 | 2.76±0.19 | 2.96±0.49 | 0.523 |
| $MLD_{after}$ [mm] | 4.97±0.27 | 4.19±0.68[a] | 4.84±0.45[b] | 0.011 | 3.52±0.31 | 3.35±0.32 | 3.65±0.46 | 0.305 |
| Overstretch ratio | 1.18±0.10 | 1.12±0.06 | 1.20±0.08 | 0.158 | 1.20±0.11 | 1.21±0.06 | 1.24±0.08 | 0.765 |

Coated, balloon coated with paclitaxel; standard PTA balloon, In.Pact Admiral; $MLD_{before}$, minimal lumen diameter before stent implantation; $MLD_{after}$, minimal lumen diameter after stent implantation followed by balloon treatment; Overstretch ratio = $MLD_{after}/MLD_{before}$.

* One vessel segment not treated by a balloon.

Data presented as mean ± SD. p-values were calculated with one-way ANOVA with post-hoc analysis (Tukey).

[a] Significant difference to HCB;

[b] significant difference to HCDCB.

Parameters which were considered to be independent on lumen diameter of vessels before stent implantation followed by balloon treatment are listed in Table 5. Parameters of QA indicating neointimal proliferation and lumen narrowing demonstrated inhibitory effects of both HCDCB and DCB. Reduction in minimal lumen diameter (MLD) at 4-week follow-up and in-stent diameter stenosis were significantly lower in HCDCB-treated vessels when compared to HCB-treated vessels (p = 0.002 and p = 0.008, respectively). No statistically significant differences between HCDCB- and DCB-groups were seen (reduction in MLD at FU: p = 0.300; in-stent diameter stenosis: p = 0.535). OCT measurements confirmed and supported the findings of QA (Table 5). The HCDCB group showed a significant reduction of neointimal thickness and area stenosis vs. the uncoated HCB group (p = 0.005 and p = 0.004, respectively), but no statistical differences to the DCB-group (p >0.999 and p = 0.850 for neointimal thickness and area stenosis, respectively). Histomorphometric parameters reflected inhibition of neointimal proliferation in the HCDCB-treated vessels, i.e., diameter stenosis and area stenosis (p = 0.038 and p = 0.058, respectively, vs. uncoated HCB-group, Table 5). There were no statistical differences in neointimal proliferation for both parameters between HCDCB- and DCB-groups (p = 0.952 for diameter stenosis and p = 0.816 for area stenosis).

**Table 4. Quantitative angiography (QA) of each vessel segment of internal iliac arteries before and after intervention.**

| | Hyper-compliant balloon | | Standard PTA balloon | p-value | Hyper-compliant balloon | | Standard PTA balloon | p-value |
|---|---|---|---|---|---|---|---|---|
| | Uncoated (HCB) | Coated (HCDCB) | Coated (DCB) | | Uncoated (HCB) | Coated (HCDCB) | Coated (DCB) | |
| | **Internal iliac proximal** | | | | **Internal iliac distal** | | | |
| n (analyzed vessels) | 8 | 8 | 8 | | 7* | 8 | 7** | |
| $MLD_{before}$ [mm] | 3.11±0.26 | 3.13±0.33 | 3.07±0.19 | 0.902 | 2.36±0.15 | 2.67±0.74 | 2.42±0.20 | 0.420 |
| $MLD_{after}$ [mm] | 3.47±0.29 | 3.63±0.22 | 3.48±0.28 | 0.439 | 2.70±0.17 | 3.10±0.74 | 2.93±0.28 | 0.303 |
| Overstretch ratio | 1.12±0.06 | 1.17±0.15 | 1.14±0.08 | 0.570 | 1.15±0.12 | 1.18±0.15 | 1.21±0.06 | 0.689 |

* One vessel segment not treated by a balloon;

** one vessel segment not evaluated due to outflow obstruction.

Data presented as mean ± SD. p-values were calculated with one-way ANOVA with post-hoc analysis (Tukey).

**Table 5. Main findings from QA, OCT and histomorphometry at 4-week follow-up.**

| | Hyper-compliant balloon | | Standard PTA balloon | p-value HCDCB vs. |
|---|---|---|---|---|
| | Uncoated (HCB) (single balloon size) | Coated (DCHCB) (single balloon size) | Coated (DCB) (3 different diameters, each adjusted to vessel segment)[a] | a) HCB<br>b) DCB |
| **Quantitative angiography (QA)** | | | | |
| **n (analyzed vessels)** | 30* | 32 | 30** | |
| **Reduction in MLD at FU [mm]** | 1.12±0.68 | 0.63±0.39 | 0.84±0.54 | a) 0.002<br>b) 0.300 |
| **Diameter stenosis at FU [%]** | 30.3±16.7 | 18.7±12.3 | 22.8±15.5 | a) 0.008<br>b) 0.535 |
| **Optical coherence tomography (OCT)** | | | | |
| **n (analyzed vessels)** | 21*¶♦ | 24¶ | 22**¶ | |
| **Neointimal thickness [mm]** | 0.31±0.17 | 0.19±0.09 | 0.21±0.11 | a) 0.005<br>b) >0.999 |
| **Area stenosis [%]** | 34.7±16.0 | 21.4±11.6 | 23.6±12.3 | a) 0.004<br>b) 0.850 |
| **Histomorphometry** | | | | |
| **n (analyzed vessels)** | 30* | 32 | 30** | |
| **Injury score** | 2.27±0.64 | 2.52±0.52 | 2.41±0.61 | a) 0.292<br>b) >0.999 |
| **Diameter stenosis [%]** | 32.5±8.5 | 27.5±7.1 | 26.9±8.0 | a) 0.038<br>b) 0.952 |
| **Area stenosis [%]** | 53.2±11.5 | 46.7±9.6 | 45.1±11.6 | a) 0.058<br>b) 0.816 |

Data obtained for parameters selected here were combined for all vessels. FU, follow-up;

[a] DCB with 4.0, 5.0 and 6.0 mm in diameter.

* One distal segment each in internal iliac and femoral arteries not treated by a balloon;

** one distal segment in femoral arteries not treated by a balloon, and one distal segment in internal iliac arteries not evaluated due to outflow obstruction;

¶ measurement in vessel segments with a large diameter in femoral arteries (here: proximal segments) not possible due to methological limitations;

♦ imaging catheter not advanceable into one vessel segment.

Data presented as mean ± SD. p-values were calculated with one-way ANOVA with post-hoc analysis (Tukey or Dunn).

For each vessel segment groups of both peripheral arteries mean values ± SD and p-values of OCT and histomorphometric data are listed in S1 and S2 Tables.

### Injury

Histological injury scores were not statistically different among the three treatment groups (p = 0.252, Table 5).

## Discussion

The superficial femoral artery (SFA), as the longest peripheral artery, is the most commonly diseased artery in the peripheral vasculature with more than 50% of all peripheral artery disease (PAD) involved [13]. Patients with SFA disease often present with long, diffuse lesions including difficulty crossing excessive long-segment occlusions of >25 cm in length, and long femoropopliteal lesions still represent one of the major challenges of endovascular therapy for PAD.

Vessel preparation using PTA with uncoated balloons, high-pressure PTA balloons or scoring and cutting balloons and/or debulking strategies such as atherectomy for luminal

gain, plaque modification, and enhancement of drug uptake especially in calcified lesions, which are prevalent in the SFA, prior to treatment with a DCB is now an integral part of restenosis prophylaxis [14, 15]. However, in order to avoid further lumen expansion by conventional balloons with defined diameters which may again results in arterial wall injury prior selection of balloons with an appropriate diameter and length from a stock of a variety of devices with different dimensions as well as precise placement of balloons is required. Especially in long-segment SFA lesions (>25 cm) with different diameters along their length more than one PTA balloon with defined diameter is usually needed to be employed to fully cover the lesion. In reference to these complications and factual evidence there is demand for a device with smooth mechanical effect that (i) adapts to the vessel structure regardless of the vessel diameters along the length of lesions due to its high conformability to vascular anatomy, (ii) and provides a balloon length for fully coverage of the lesion to be treated in one go (e.g. 20–40 cm).

CE-marked hyper-compliant (elastomeric) balloons have been used for temporary occlusion of perforated artery to restore hemostasis [16], additionally for balloon-assisted coil embolization in the endovascular treatment of aneurysms [17], or to facilitate endograft or stent placement in complicated vessel anatomy where optimal sealing (often resulting in endoleaks) and fixation have proven difficult [18]. These balloons are usually short in length (up to 30 mm), typically made of polyurethane or silicone, and are able to stretch 100% to 800%.

Hyper-compliant balloons accommodate a wide range of vessel diameters and adjust the shape of the artery in which they are inflated. Inflation is volume-driven rather than pressure-driven, and once the balloon is circumferentially in contact with the vascular structure it elongates to its entire length without undue radial force, thereby preventing trauma to the vessel wall. Also, it is suggested that hyper-compliant balloons are more useable in the treatment of bifurcation than PTA balloons which maintain a constant shape (i.e., cylindrical shape) with increased pressure. The suppleness of the balloon membrane allows hyper-compliant balloons to form nodes into arterial branches upon inflation. Due to the absence of compliance (conformability) PTA balloons do not have the propensity to adapt to the anatomy of arterial bifurcation.

To our best knowledge there are no hyper-compliant balloon catheters being coated with a drug available on the market. Especially for the drug transfer, irrespective of the vessel diameter and/ or state of the vascular wall (i.e., smooth or irregular) a full contact with the vessel wall over the entire length of the vessel segment to be treated may be achieved with hyper-compliant drug-coated balloons. Exact vessel size measurement is not necessary anymore, avoiding the use of multiple PTA balloons and/or under- or oversizing of the PTA balloon.

In the present study we investigated whether one single long hyper-compliant drug-coated balloon (HCDCB) may be an alternative to the use of two different semi-compliant DCB, each selected according to the respective vessel diameter. For this purpose, two suitable vessel segments of each internal iliac and femoral artery, differing by 1–2 mm in diameter from each other, were subjected to stent implantation followed by balloon treatment.

A stent overstretch model was used to provoke neointimal thickening and to define an initial vessel lumen and luminal surface [19]. The effect of HCDCB on neointimal formation was compared with a clinically proven peripheral DCB (In.Pact Admiral) [3, 20]. The experiments on the two different coated balloon constructs (HCDCB and DCB) were controlled by comparison to uncoated hyper-compliant balloons (HCB).

Radiographs showed that at low pressure (< 2 atm) long HCDCB (length: 100 or 150 mm fitting to the vessel length in pigs) accomodate the vessel structure with different vessel diameters ranging between 2.24 mm (minimum) and 5.67 mm (maximum) as measured for minimal

lumen diameter (MLD) after treatment. Further, they adjusted to the endoluminal surface of the artery with circumferential apposition, providing the prerequisites for an optimal vessel wall contact for drug delivery with HCDCB.

The present study confirmed the findings made in the previous study [10] demonstrating that both HCDCB and DCB are similarly effective in inhibiting neointimal proliferation. Furthermore, it highlighted the ability of one single long HCDCB to suppress neointimal thickening in segments differing in vessel diameters in one step.

Further, a number of parameters supports the safety and efficacy of HCDCB, i.e. high vessel patency at 4-week follow-up, no thrombi detected on the balloon catheters directly after intervention, no animal deaths or clinical symptoms during 4 weeks of follow-up.

The present work suggested that the use of one single long HCDCB allows for treatment of more than one lesion and/ or stented vessel segment with different vessel diameters in an artery, replacing treatment with DCB with a less extensible diameter, each selected according to the dimensions of the treatment site.

## Limitations

The main limitation of this study derives from its experimental nature. As usual in preclinical studies number of large animals was limited and this study used young animals with vessels lacking typical types of human vascular disease in superficial femoral artery (i.e., long lesions being either stenotic or occluded with high calcific burden). Also, the length of the SFA in pigs does not correspond to that in humans. Only balloons of shorter lengths (up to 15 cm) could be employed in femoropopliteal arteries of young animals (about 3 months old) compared to those when intended to be used to treat femoropopliteal lesions in humans (e.g. 30–40 cm long balloons). This is due to the size and anatomy of the experimental animals. Thus, as already stated in [10], the results presented in this study may not reflect the findings in patients with peripheral arterial disease. Also, sustainability of treatment effects with hyper-compliant drug-coated balloons (HCDCB) cannot be evaluated in the existing animal model, a known shortage of the model [21]. Only a clinical trial with hyper-compliant balloons can provide information on efficacy and potential safety risks in patients.

## Conclusions

Hyper-compliant drug-coated balloons (HCDCB) inhibited injury-induced excessive neotininal proliferation at low inflation pressure with an efficacy similar to that of clinically proven DCB in the animal model. High conformability and long length of HCDCB may offer advantages over less vessel-adapted DCB in reducing the number of devices to be used for treatment of long diffusely diseased vessels often characterized by different diameters and irregularities of the vessel shape, resulting in a less time-consuming and more cost-effective treatment. Due to high adaptance of the balloon to the vessel anatomy at low inflation pressure as observed in porcine peripheral arteries long HCDCB may also accommodate to different diameters of long lesions in humans. Since HCDCB are not able to dilate the stenotic artery or crack the calcified plaques (they are not intended for this purpose) they may be applied after lesion preparation. Besides that, it is known that calcium is a physical barrier to optimal drug transfer to the vessel wall and vessel wall distribution, especially in circumferential calcification [15]. Therefore, the procedure involving HCDCB is first only used in cases of low to moderate calcification or as mentioned above after plaque incision or debulking (i.e., after vessel preparation). Finally, this question has to be answered by clinical studies.

## Supporting information

**S1 Table. OCT data for each group of vessel segments in peripheral arteries at 4-week follow-up.**
(DOCX)

**S2 Table. Histomorphometric data for each group of vessel segments in peripheral arteries at 4-week follow-up.**
(DOCX)

**S1 Checklist. The ARRIVE guidelines 2.0: Author checklist.**
(PDF)

## Acknowledgments

The authors specially thank Dr. M. Löchel for coating the hyper-compliant balloons, Dr. D. Schütt for performing paclitaxel analysis, and A. Mittag for conducting interventions.

## Author Contributions

**Conceptualization:** Stephanie Bienek, Maciej Kusmierczuk, Stephanie Bettink, Bruno Scheller.

**Data curation:** Stephanie Bienek, Maciej Kusmierczuk, Beatrix Schnorr, Stephanie Bettink.

**Formal analysis:** Stephanie Bienek, Maciej Kusmierczuk, Stephanie Bettink, Bruno Scheller.

**Funding acquisition:** Bruno Scheller.

**Investigation:** Stephanie Bienek, Maciej Kusmierczuk, Beatrix Schnorr, Ole Gemeinhardt, Bruno Scheller.

**Methodology:** Stephanie Bienek, Maciej Kusmierczuk, Stephanie Bettink, Bruno Scheller.

**Project administration:** Stephanie Bienek.

**Resources:** Bruno Scheller.

**Validation:** Maciej Kusmierczuk, Beatrix Schnorr, Stephanie Bettink.

**Visualization:** Bruno Scheller.

**Writing – original draft:** Stephanie Bienek, Bruno Scheller.

**Writing – review & editing:** Maciej Kusmierczuk, Beatrix Schnorr, Stephanie Bettink.

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
