## [Decision Letter · Decision Letter 0]

10 Oct 2022

PONE-D-22-23006One single drug-coated balloon for all shapes/diameters? Neointimal proliferation inhibition in porcine peripheral arteriesPLOS ONE

Dear Dr. Scheller,

Thank you for submitting your manuscript to PLOS ONE. After careful consideration, we feel that it has merit but does not fully meet PLOS ONE’s publication criteria as it currently stands. Therefore, we invite you to submit a revised version of the manuscript that addresses the points raised during the review process.

We look forward to receiving your revised manuscript.

Kind regards,

Athanasios Saratzis

Academic Editor

PLOS ONE

Journal Requirements:

2. As part of your revision, please complete and submit a copy of the Full ARRIVE 2.0 Guidelines checklist, a document that aims to improve experimental reporting and reproducibility of animal studies for purposes of post-publication data analysis and reproducibility: https://arriveguidelines.org/sites/arrive/files/documents/Author%20Checklist%20-%20Full.pdf  (PDF). Please include your completed checklist as a Supporting Information file. Note that if your paper is accepted for publication, this checklist will be published as part of your article.

"Bruno Scheller is a shareholder of InnoRa GmbH."

Additional Editor Comments:

Can the Authors please provide some discussion with regards to future plans and methodology for human testing? Please refer to Reviewer 1 comments.

Reviewers' comments:

Reviewer's Responses to Questions

**Comments to the Author**

1. Is the manuscript technically sound, and do the data support the conclusions?

Reviewer #1: Yes

Reviewer #2: Yes

2. Has the statistical analysis been performed appropriately and rigorously? 

Reviewer #1: Yes

Reviewer #2: Yes

3. Have the authors made all data underlying the findings in their manuscript fully available?

Reviewer #1: Yes

Reviewer #2: Yes

4. Is the manuscript presented in an intelligible fashion and written in standard English?

Reviewer #1: Yes

Reviewer #2: Yes

5. Review Comments to the Author

Reviewer #1: This is a well written and well conducted experimental project. The idea of long hyper-compliant balloons for drug delivery in humans as very appealing and makes logical sense. High pressure modelling is not required for drug-delivery and instead excellent vessel wall apposition is key. This can potentially be achieved with hyper-compliance balloons.

The balloon lengths used in the animals studied were 10-15cm, but in humans, one would hope that 30-40cm balloons would be feasible in order to treat the entire popliteal and superficial femoral arterial section in one go. It would be good for the authors to discuss this in the their conclusions and the plan for in human testing.

Reviewer #2: I would like to congratulate the authors for their study. Development of new devices for the treatment of peripheral arterial disease is of paramount importance. The text is well written, comprehensive without leaving any question to the scientific community. However application of such device in humans is still under investigation especially in patients with long lesions which are either stenotic or occluded with high calcific burden.

Do the authors believe that HCDCB are effective in calcified lesions where drug delivery across the arterial wall is low?

Do the authors believe that HCDCB at low pressures can fully accommodate to different diameter lesions in humans?

Pg3 line 62 Please correct vessles to vessels

6. PLOS authors have the option to publish the peer review history of their article (what does this mean?). If published, this will include your full peer review and any attached files.

Reviewer #1: **Yes: **Dominic PJ Howard

Reviewer #2: No

---

## [Author Response · Author response to Decision Letter 0]

28 Nov 2022

Rebuttal letter to manuscript PONE-D-22-23006

I) Editor Comments

Can the authors please provide some discussion with regards to future plans and methodology for human testing? Please refer to Reviewer 1 comments.

Our response

Possible clinical scenarios are discussed in the Conclusions section.

II) Reviewer #1

… The balloon lengths used in the animal study were 10-15 cm, but in humans, one would hope that 30-40 cm balloons would be feasible in order to treat the entire popliteal and superficial femoral arterial section in one go. It would be good for the authors to discuss in their conclusions and the plan for in human testing. 

Our response

We have addressed the reviewer’s #1 view regarding the balloon lengths used in our animal study by stating that the use of balloons of shorter lengths (i.e., 10-15 cm) in our animal study is related to the size and anatomy of the experimental animals (see section ‘Limitations’, lines 427-431). 

III) Reviewer #2 

… However application of such device in humans is still under investigation especially in patients with long lesions which are either stenotic of occluded with high calcific burden.

a) Do the authors believe that HCDCB are effective in calcified lesions where drug delivery across the arterial wall is low?

Our response

We have added a statement concerning calcium being a barrier to drug transfer and vessel wall distribution, and a corresponding citation (Fanelli et al., 2014 [15]) (see section ‘Conclusions’, lines 448-450).

b) Do the authors believe that HCDCB at low pressures can fully accommodate to different diameter lesion in humans?

Our response

We have observed in our several animal experiments that at low inflation pressure long hyper-compliant balloons (HCDCB) well adapt to the vessel structure regardless of vessel diameters, also by forming protrusions at vessel branches. A statement addressing this issue is read in the section ‘Conclusions’. 

c) Pg3 line 63 Please correct vessles to vessels

Our response

 This correction has been made.

---

## [Editor Report · Decision Letter 1]

22 Dec 2022

One single drug-coated balloon for all shapes/diameters? Neointimal proliferation inhibition in porcine peripheral arteries

PONE-D-22-23006R1

Dear Dr. Scheller,

We’re pleased to inform you that your manuscript has been judged scientifically suitable for publication and will be formally accepted for publication once it meets all outstanding technical requirements.

Kind regards,

Athanasios Saratzis

Academic Editor

PLOS ONE
---

## [Editor Report · Acceptance letter]

19 Jan 2023

PONE-D-22-23006R1 

One single drug-coated balloon for all shapes/diameters? Neointimal proliferation inhibition in porcine peripheral arteries 

Dear Dr. Scheller:

I'm pleased to inform you that your manuscript has been deemed suitable for publication in PLOS ONE. Congratulations! Your manuscript is now with our production department. 

Kind regards, 

on behalf of

Dr. Athanasios Saratzis 

Academic Editor

PLOS ONE